# Thermal Probing Techniques for a Single Live Cell

**DOI:** 10.3390/s22145093

**Published:** 2022-07-07

**Authors:** Nana Yang, Jingjing Xu, Fan Wang, Fan Yang, Danhong Han, Shengyong Xu

**Affiliations:** 1School of Microelectronics, Shandong University, Jinan 250100, China; yangnana@pku.edu.cn (N.Y.); wff@mail.sdu.edu.cn (F.W.); 2School of Electronics, Peking University, Beijing 100871, China; fyang1992@pku.edu.cn (F.Y.); handanhong@126.com (D.H.); xusy@pku.edu.cn (S.X.); 3Beijing Research Institute of Mechanical Equipment, Beijing 100854, China

**Keywords:** cell temperature, temperature sensor, fluorescence thermometry, thin-film thermocouple, multiscale measurement system

## Abstract

Temperature is a significant factor in determining and characterizing cellular metabolism and other biochemical activities. In this study, we provide a brief overview of two important technologies used to monitor the local temperatures of individual living cells: fluorescence nano-thermometry and an array of micro-/nano-sized thin-film thermocouples. We explain some key technical issues that must be addressed and optimised for further practical applications, such as in cell biology, drug selection, and novel antitumor therapy. We also offer a method for combining them into a hybrid measuring system.

## 1. Introduction

Cells are basic units of an advanced biosystem. However, certain essential events occurring in the small area of a single cell still remain unknown, such as details about the exchange of material and energy in the metabolism, regulations, and mechanisms in coding and expressing genes, proteins, enzymes, and DNA synthesis.

With a diameter of around ten microns, a cell is so small that it requires probes at the micro- and nano-scales in order for researchers to obtain the physical, chemical, and biological information inside a cell. Electro-probes, for example, are utilized to obtain neuro-signals and transmembrane ion channels [1,2]. Optical techniques have been used to obtain the morphology and location of sub-cell organelles, nuclei, chromosomes, and even enzymes and proteins [3,4].

Temperature is an important natural parameter of a biosystem. It represents the average kinetic energy of molecules, ions, and atoms. Consequently, temperature is one of the most essential criteria for determining and characterizing a cell’s life conditions [5,6]. For instance, cellular pathogenesis is characterized by aberrant heat generation. During the last two decades, fluorescence thermometry has rapidly developed as a method for monitoring temperature in individual live cells [7,8,9]. Fluorescence thermometry used temperature-sensitive nanomaterials and molecules as temperature indicators, and advanced optical equipment, such as fluorescence microscope and confocal laser scanning microscope (CLSM), was used to map the fluorescence spectra within a single cell. Then, the local temperature values were obtained by comparing the recorded fluorescence spectra to the pre-calibrated data under similar or the same environmental conditions at set temperatures [10,11,12,13,14,15,16]. Many intriguing findings have been published, for example, the temperature of a cell’s nucleus was discovered to be greater than the average temperature of the rest of the cell [17,18,19].

To monitor the local temperature of a single cell, an alternative approach that has emerged in recent years is thin-film-based micro-/nano-scaled thermocouple (TC), as well as TC arrays [20,21]. A thermocouple is usually composed of two electrical conductors with different Seebeck coefficients, serving as a thermometer, and is based on the thermoelectric principle. It produces temperature-dependent voltage as a result of the Seebeck effect, which is the electromotive force that develops across two points of an electrically conducting material when there is a temperature difference between them [22]. Thermocouples are widely used as temperature sensors, because the voltage can be interpreted to measure temperature. A TC is a passive sensor that does not add any heat to the testing area. It is also a high-resolution, reliable, and fast-responding sensor. A TC might be built on the tip of an atomic force microscope (AFM) [23] using typical manufacturing procedures for micro-/nano-devices in the semiconductor industry in order to measure the thermal activities of a single cell [24]. Nano-tip-shaped W-Pt TCs have been made using electrochemical techniques to penetrate a live cell and to monitor the temperature change under external chemicals [20,25]. Freestanding micro-probes with built-in TC sensors have been fabricated on Si_3_N_4_ tips and have been used to monitor the local temperature inside a neuron cell, where a transient pulse of 7.5  K rise was recorded in some cell processes [26]. These customized TCs, however, are not ideal for continuous, long-period observation.

Recently, the minimum width of a TC was reduced to less than 200 nm [27,28,29,30]. The thermal noise of the measurement system at room temperature was reduced to ±5 mK. A freestanding Si_3_N_4_ thin-film base was used to reduce the dissipation and improve sensitivity [31]. Several general thermal properties of cultivated HeLa were observed over days [32]. However, such a multiscale measuring system requires more development optimization.

In this short review, we briefly comment on the potential of a hybrid approach for probing the local temperatures of a single cell by combining the fluorescence nano-thermometer and thin-film TC arrays. We also highlighted numerous critical difficulties that must be addressed in this hybrid technique.

## 2. Typical Results from Fluorescence Probing of a Single Cell

The technique of fluorescence nano-thermometry could be traced back to the 1940s [33], which was quickly developed and applied to detect the thermal properties of cells [19,34,35,36,37]. The thermal resolution of organic fluorophores was too poor and could not identify a difference of 1 °C due to the fluorophores’ low sensitivity to temperature fluctuation. Thus, in 2003, the Uchiyama group attempted to separate the requisite roles of fluorescent molecular thermometers, i.e., sensing temperature fluctuation [38].

Since then, more than 45 groups have reported over 60 fluorescence probes, through the use of various environment-sensitive fluorophores, including dansylamine [39], naphthalimide [40,41], and 7-nitro-2,1,3-benzoxadiazole (NBD) [42,43,44]. Furthermore, viscosity-sensitive fluorophores have been proposed as distinct fluorescent units in fluorescent thermometers [45,46].

In terms of temperature-sensitive nano-indicators, fluorescence nano-thermometry can be catalogued with different materials, including polymer molecules [34,47], quantum dots [48,49], organic clusters [50], bio-molecules [51,52], and DNA sectors [53,54]. In terms of the sensing mechanism and calibration techniques, they can be classified into four classes, namely those that use signal intensity [15,16,55], lifespan [13,14], excitation wavelength polarization [10], and spectrum [8] as the essential parameters.

To give a glimpse of the state-of-the-art performance of fluorescence nano-thermometers, here, we briefly introduce two pieces of typical and important work that have revealed unknown facts about cell biology.

### 2.1. A Cell Nuclei Has a Higher Temperature than the Surrounding Plasma

In 2015, Hayashi et al. reported an important result; they found that the temperature of the cultured cell nuclei was higher than the surrounding plasma by an average value of nearly 1.0 °C [17]. This was not a surprising discovery, because the temperature at the heat source (the nuclei) must be higher than average (the surroundings) according to thermodynamics. However, this was the first time it had been validated by direct experimental measurements. They created an autonomously cell-permeable fluorescent polymeric thermometer (FPT), which solved the technical challenge of FPT being injected into mammalian cells by microinjection that was present in prior studies [56]. As shown in Figure 1a, the cell-permeable FPT is a combination of a thermosensitive unit (N-n-propylacrylamide (NNPAM)), a cation unit beneficial for cell autonomous penetration (3-(acrylamidopropyl)trimethylammonium (APTMA)) [56], and a fluorescent unit (N-{2-[(7-N,Ndimethylaminosulfonyl)-2,1,3-benzoxadiazol-4-yl](methyl)amino}ethyl-N-methylacrylamide (DBThD-AA)), which displayed a temperature resolution of 0.05–0.54 °C within 28–38 °C in HeLa cells. After being injected into HeLa cells for 10 min, the cell-permeable FPT spontaneously diffused throughout the cells, including the cytosol and nucleus (Figure 1b). The FLIM analysis for the FPT fluorescence lifetime in 49 cell samples indicated an average temperature difference of 0.98 °C between the nucleus and the cytoplasm (Figure 1c,d).

### 2.2. Local Temperature of Mitochondria

The effect of heat production in the mitochondria and other cellular components is vital to understanding their roles in the various metabolic activities of cells. Arai et al. developed Mito Thermo Yellow (MTY), a small molecule fluorescent thermometer with a mitochondrial targeting capacity, in 2015, with the ability to detect the intracellular temperature gradient in diverse cells [57]. Figure 2a shows the chemical structure of the MTY fluorescent probe, which has a temperature resolution of 1.12 °C. Consequently, MTY, whose fluorescence intensity declines with temperature, has been proven to be useful for mapping the distribution of the intercellular temperature. P. Rustin’s team studied mitochondrial thermogenesis in human embryonic kidney (HEK) 293 cells and primary skin fibroblasts in 2018 by monitoring oxygen consumption (or stress) and changes in MTY fluorescence at the same time [58]. Adherent HEK293 cells were kept at 38 °C for 10 min before being added to an oxygen-rich buffer. Then, as shown in Figure 2b, they immediately started to consume oxygen, highlighted with red trace, accompanied by a progressive decrease in MTY fluorescence (blue trace; phase I) to a stable minimum (phase II). Because of the depletion of all of the oxygen in the cuvette, the directional shift of MTY fluorescence reversed and was progressively recovered practically to the beginning value in phases III and IV (red trace). To calibrate the fluorescence signal, the medium temperature was gradually increased (green trace), followed by a consistent change in MTY fluorescence (phase VI). This supported the validity of the MTY thermometric performance. The rise in mitochondrial temperature caused by the complete activation of respiration was calculated to be around 10 °C over the whole test. This suggests that the temperature of the mitochondria in the human body may reach close to 50 °C with normal respiration.

Because of the length of this brief review, many other interesting results have not described here. Okabe et al., for example, detected a periodical shift in temperature in COS7 cells by utilizing continuous monitoring with fluorescent nano-thermometry [59]. Fluorescence nano-thermometry is still a hot subject, and new nano-indicators are constantly being produced [60,61]. For more details, there are some excellent review papers regarding the devilment of fluorescence nano-thermometry [62,63,64,65,66].

## 3. Technical Issues in Fluorescence Nano-Thermometer

Fluorescence nano-thermometers have been well developed for detecting the temperature distribution in a single cell. They might provide a two-dimensional (2D) map of the temperature distribution in a single cell as an outstanding feature. These strategies, however, have certain drawbacks. For example, in one experiment, the temperature value was derived by comparing it with the pre-calibration data. However, the true environment in a cell is never the same as that utilized in the calibration operations in terms of pH, concentrations of different ions and proteins, and so on [67,68,69,70,71]. This raised arguments about the reported experimental data [72,73,74].

Besides the influence of the local environment, which results in a relatively poor temperature resolution and large measurement error, the local heating effect caused by excitation lights, uncertainty in the endocytosis of the cell membrane, and a small working distance, are three of the main issues being faced regarding the further development of these techniques.

The heating effect is an intrinsic drawback of the fluorescence thermal probing techniques. To record the fluorescence spectra of nano-sized sensitive objects in a cell, a high incidence external light source with a relatively high incident light flux is required as an excitation source. This external light source increases the absolute local temperature of the cell membrane, nuclei, organelle, and cytoplasm. Meanwhile, the difference in the adsorption of lights creates measurement errors in the temperature distribution inside a cell. To limit the measurement error, a cool light source can be utilized and thermal probing can be conducted in a discontinuous way; that is, measure for a brief period and then switch off all external lights for a longer duration to allow the system to return to its normal state.

The choice of nano-sized temperature-sensitive materials, i.e., thermal indicators, is critical for nano-fluorescence thermal probing techniques. Nanomaterials enter cells via the diffusion or endocytosis processes. However, these processes, particularly the endocytosis effect, are not always under control, and they can alter the natural status of live cells. Furthermore, once inside the cell, they may not attach to the target positions of the cell organelle as intended. For the specific purpose of single cell experiments, the condition must be tuned through the proper selection of one or two nano-thermal indicators.

## 4. Main Issues of the Micro-/Nano-Thin-Film Thermocouple Approach

Thin-film thermocouples (TFTCs) have the advantage of a passive sensing mechanism that does not generate additional heat in the test region. TFTCs can be fabricated with standard cleanroom techniques and processes that are compatible with those for integrated circuits, giving them the advantages of simple fabrication for mass production and a size than it scalable down to micrometer and nanometer scales.

Similar to a commercial thermocouple that is made with two different metallic wires and works based on the Seebeck effect, a TFTCs is made with two different metallic thin-film stripes, A and B, and also works based on the same Seebeck effect, as shown in Figure 3 [75,76]. A thermoelectric voltage linearly proportional to the temperature difference between the hot and cold ends is measured as ΔV = (S_A_ − S_B_)·(Th − Tc), where S_A_ and S_B_ represent the Seebeck coefficients of thin-film stripes A and B, respectively, and Th and Tc are the temperatures of the hot and cold ends, respectively [77]. In a narrow temperature range, S_A_ and S_B_ are usually constants, so the thermopower or sensitivity of the TFTC, defined as S = S_A_ − S_B_, is also a constant. This leads to ΔV = S·ΔT, where ΔT = (Th − Tc), showing a unique advantage of simplicity for real-time measurement when compared with the complicated calibration processes of various fluorescence thermometry techniques. Meanwhile, TFTCs offer a reliable resolution of 10–50 mK at room temperature, depending on the setup of the measurement environment.

For the application of TFTCs in probing the local temperature of a single cell, several technique issues need to be addressed.

### 4.1. Minimization of the TFTC Devices and Related Scale Effects

A traditional TC is made of two metallic wires with very different Seebeck coefficients, with wire diameter diameters ranging from 0.1 mm to 1.0 mm. A cell is typically 5–20 microns in size. Therefore, to measure the local temperature of a single cell, the junction region, the hot end, should be minimized to several microns or less.

Liu et al. studied various TFTCs made with Cr and Ni thin-film stripes. The results showed that the sensitivities of the TFTCs were not sensitive to the junction size from 20 × 20 μm^2^ down to 3 × 3 μm^2^ on both glass and silicon wafers, which remained around 26.2 ± 1.5 μV/K. Nonetheless, the absolute Seebeck coefficients of the metallic thin films were found to be smaller than those of bulks of the same materials. The Cr–Ni micro-TFTCs, for instance, repeatedly showed a sensitivity of 26–27 μV/K, smaller than the theoretically expected 36.8 μV/K [77].

This size effect of the Seebeck coefficient on the width of the thin-film stripe has been extensively studied [27,28,29,79,80,81,82,83]. It demonstrated that decreasing the stripe width from the micron to the submicron and nanometer scales reduced the sensitivity of TFTCs even further. Yet, even with a small sensitivity of 1–2 μV/K, the thermocouple was still able to offer a temperature resolution of 20–50 mK.

Huo et al. developed a 3D stacking process to fabricate very small TFTCs out of Cr, Au, and Pd thin-film stripes. A junction width of 138 nm was achieved [30]. The intrinsic heat capacity of such a nano-scale sensor is in the order of 10^−14^ J/K per micron length. The thermopower of the sensors was 9.6 ± 0.7 μV/K for the Cr/Pd sensors and 3.6 ± 0.1 μV/K for the Au/Pd sensors, showing measurement accuracy in the range of 40–100 mK.

Theoretically, the junction width of a TFTC could be reduced to 100 nm or less. However, this may be accompanied by increased fabrication difficulty and decreased reproducibility and stability of the sensing performance.

For an array of TFTCs with a junction width of 200–500 nm, a complicated electron beam lithography system and related thin-film techniques are required. Figure 4 presents scanning electron microscope (SEM) micrographs taken from a sample of 100 Pd–Cr submicron TFTCs fabricated on a SiO_2_/Si wafer, where (a) is the top view of the device, showing the central measurement region and 200 leads, and (b) is the central region of 100 Pd-Cr sensors arranged in a 10 × 10 array [84].

Figure 4 shows a TFTC array with 100 sensors and 200 leads. Han et al. presented a work in which the number of leads could be reduced by nearly half, without causing a significant impact on the 2D mapping of the local temperature distribution [85]. In this case, the number of leads could be reduced to 101, simplifying the device structure and suppressing the heat dissipation through metallic leads. The principle for saving half of the leads of a TFTC array is schematically illustrated in Figure 5.

### 4.2. Freestanding Platform of Si_3_N_4_ Thin-Film Window for TFTC Arrays

Heat dissipation through the substrate of TFTCs has been identified as a serious problem in the thermal probing of local temperature at micro-/nano-scales [31]. For example, in a transmission electron microscope (TEM) or SEM, a focused electron beam (e-beam) were used to melt metallic thin films or nanowires, or could even vaporize parts of them [86,87], indicating that the local temperature under the focal point of the incident e-beam was higher than the melting points or even vaporization points. However, when a micro-TFTC made on a SiO_2_/Si wafer was placed at the focal point of the incident e-beam, it recorded a much lower temperature.

In sharp contrast, if the same TFTC was made on a 400 nm thick, freestanding Si_3_N_4_ thin-film, and the same measurement was repeated, the local temperature increment was 30 times higher than that on the Si wafer. Similarly, a local temperature increment peak was recorded at a focal point of an incident laser beam up to 2400 K, which was 100 times higher than that measured with the same sensors made on thick Si wafers [88]. The large difference measured in the sensitivity of the same TFTC sensors was attributed to the dissipation effect of heat. On the 400 nm thick freestanding Si_3_N_4_ window, the heat dissipation at the focal point region was much smaller than that on the 400-micron-thick SiO_2_/Si wafer.

The development of a freestanding Si_3_N_4_ thin-film window platform significantly improved the thermal sensitivity of the built-in micro-TC arrays for sensing individual cells. Recently, Han et al. applied such a platform and observed local temperature increments of individual HeLa cells in the range of 10–50 mK [88].

Si_3_N_4_ thin-films have superior mechanical strength. For example, these films are applied to construct a liquid sample cell inside a TEM for studies on bio-samples under live conditions [89,90]. For a freestanding window with an area of 1.0 × 1.0 mm^2^ in the thermal probing of a single cell, the film thickness could be reduced from 400 nm to 40–100 nm, further reducing the heat dissipation and improving the sensitivity of the TFTC sensors built on the thin-film platform.

### 4.3. Measurement Circuit and Data Processing Software

When sensing the temperature of a single cell, the micro-/nano-TFTC produces a DC voltage in the sub-microvolt to a few microvolt range [21,88], so the measurement system requires a nanovoltmeter. A multiplexer that offers an opening operation through measurement of the sensors on a time scale in a controlled manner is required for multiple sensors, such as an array of 100 TFTC sensors [77,81]. As the measured data are low DC voltages, electric relays can be used as switches for individual channels.

Liu et al. developed a strategy for 2D mapping of the local temperature after recording the data from all of the sensors for a period of time [77], where a limited number of sensors could result in a smooth 2D map of the local temperatures of the testing region at any given time. Later, Li et al. improved on this strategy, achieving near-real-time 2D mapping of the local temperatures of the region under test with a delay of only a few seconds [81]. This is very helpful for monitoring the temperature of a single cell, as it has been shown that a cell changes its temperature at a slow rate of tens of minutes [21,88]. A near-real-time 2D map of the temperatures within a single cell, for example, may provide valuable clues for understanding the dynamic biochemical reactions inside the cell, or may provide criteria for modifying the parameters of an ongoing experiment. Such a tool has the potential to be combined with artificial intelligence (AI) for automatic control in a wide range of practical applications.

### 4.4. Thermal Noise of the Measurement System

Noise is always a serious problem in the measurement of weak signals. In experiments for sensing the local temperature of a single cell, thermal noise from the environment, electrical noise from the measurement instruments and circuits, and optical noise from external excitation lights and environmental sources must all be reduced.

Yang et al. discovered a way to reduce the influence of environmental thermal noise by 4.8 times [21]. A dual-stage system was used to accomplish this, in which the cultured cells, testing chip, and circuit, as well as the multiplexer and nanovoltmeter, were all placed inside an incubator with a constant set temperature point. Meanwhile, the incubator, gas tank, gas lines, etc., were all located in a larger container with a constant temperature. This approach suppressed the heat noise in a laboratory from ±24 mK down to ±5 mK. Indeed, the system could be further optimized for lower thermal noise in the environment.

Yang et al. designed a large cylindroid tube containing a large volume of culture medium in order to observe and record the morphology and temperature of individual cells under stable, constant environmental conditions for up to 24 h [21]. However, for a continuous experiment lasting for days or even weeks, it is preferable to change the culture medium and drugs on-site, i.e., without opening the door of the incubator. Various optical measurements must also be performed using external operations without opening the incubator.

### 4.5. Other Issues

A short working distance is also an issue. The working distance between the subject lens and the target sample is usually in the order of 1 mm so as to clearly identify the special shapes of the organelles in a cell. To monitor the thermal properties of live cells, however, a large volume of liquid culture medium for the cells must be prepared. Therefore, optical systems with an inverted microscope are frequently used.

In addition, other technical issues may arise. Some unique adhesive cells, for example, prefer one type of ship surface over another. The surface of the sensors and substrate can then be covered with a proper oxide layer of a few nanometers in thickness. In other cases, modifying these surfaces with a molecular layer of organic material may be the best option [88].

Recycling the costly testing chip is a good option. By using a proper rinsing process, it is feasible to totally remove the adhesive cells growing on the testing chip, even after many hours [88]. Accordingly, the device can be used for multiple runs of experiments, particularly with the same types of culture cells.

## 5. Conclusions

In summary, we reviewed several crucial issues in current thermal probing techniques for detecting the local temperature distribution of a cultured single live cell and its subcellular organelles. As the most popular approach, fluorescence nano-thermometry has its unique merits, including a special resolution down to the 100 nm scale, full-image of two-dimensional temperature distribution, capability for showing individual temperatures of subcellular organelles, flexible choices of nano-sized thermal sensitive fluorescence materials, etc. Yet, the fluorescence nano-thermometry has its limitations. The temperature data are calculated from the measured spectra by a calibration process. In addition to the influence of the measurement environment in a cell and the difficulty in the calibration of the thermometer, we have pointed out several other critical technical challenges. For example, in long-time experiments for observing optical images and recording fluorescence spectra, the external photon flux lead to a remarkable heating effect to the target cell.

In studies on thermal probing of a single live cell, the development of contact thermal sensors, such as the micro-nano thermometry based on thin-film TC arrays, provided a complementary approach to fluorescence nano-thermometry. Thermocouples can continually and reliably measure the real-time local temperature of a single cell for up to days, at a resolution of ±10 mK. Meanwhile, they add no external heat to the system, with no effect on the thermal status of the cells under test. The development of the freestanding Si_3_N_4_ thin-film window improved the measurement platform even further, significantly increasing the thermal sensitivity of the micro-nano-TC arrays built on it.

Therefore, we conclude that a hybrid thermal probing system, comprised of a freestanding platform with a thin-film nano-thermocouple array and an optical platform of fluorescence nano-thermometry, is the optimized system for monitoring the temperature changes in individual cultured cells. Such a system can be used to measure the absolute local temperature of the target cell under test over long periods of time, up to days and weeks, and to map the temperature distribution inside the cell at the submicron scale on occasion. The results of such a hybrid system may provide unique and complete sets of data revealing previously unknown aspects of the cell cycling mechanism. The data may also reveal detailed tumour cell reactions to various drugs, as well as to chemical and physical treatments of novel therapies medicine.

## Figures and Tables

**Figure 1 sensors-22-05093-f001:**
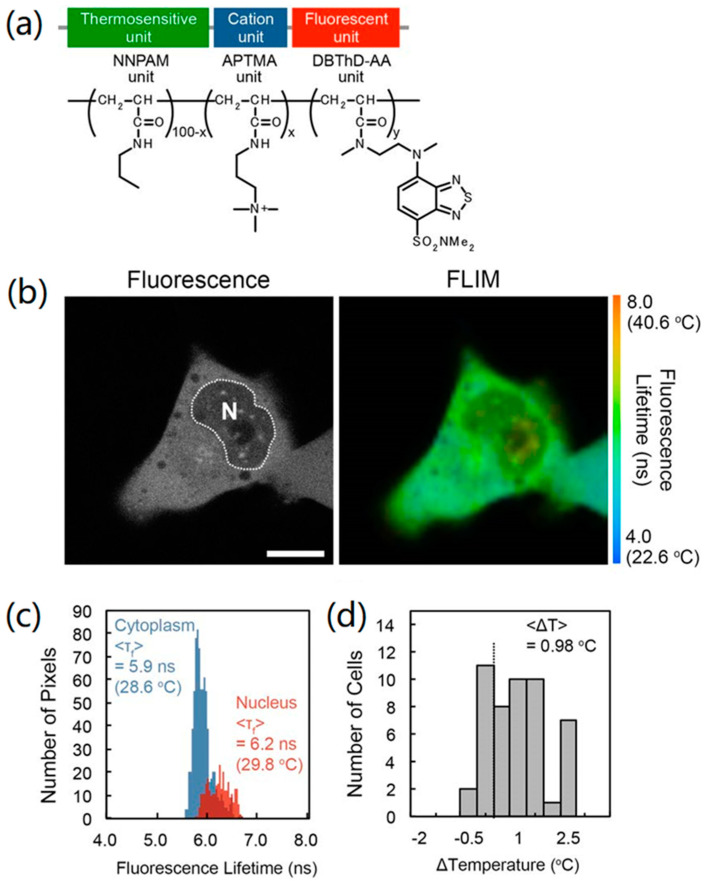
The chemical components and temperature mapping of living HeLa cells of the cell-permeable FPT. (**a**) Chemical structure of the FPT. (**b**) Fluorescence lifetime pictures and confocal fluorescence images of the FPT in a HeLa cell. N stands for nucleus. (**c**) Histograms of the fluorescence lifetime in the cell nucleus (red color) and the cytoplasm (blue color) in the cell in (**b**). (**d**) Histogram of the temperature difference between the cell nucleus and the cytoplasm (n = 49). Scale bar = 10 µm [17].

**Figure 2 sensors-22-05093-f002:**
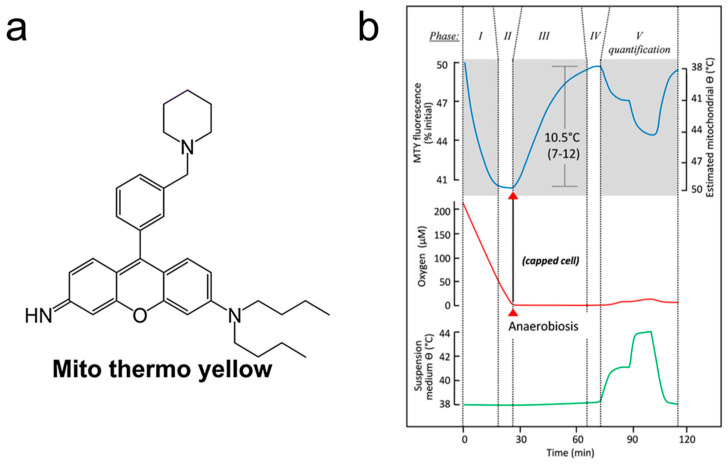
Mito Thermo Yellow (MTY). (**a**) Chemical structure of the MTY fluorescent probe [57]. (**b**) Determination of mitochondrial temperature in MTY-preloaded HEK293 cells. Phase I–phase IV support the validity of the MTY thermometric performance [58].

**Figure 3 sensors-22-05093-f003:**
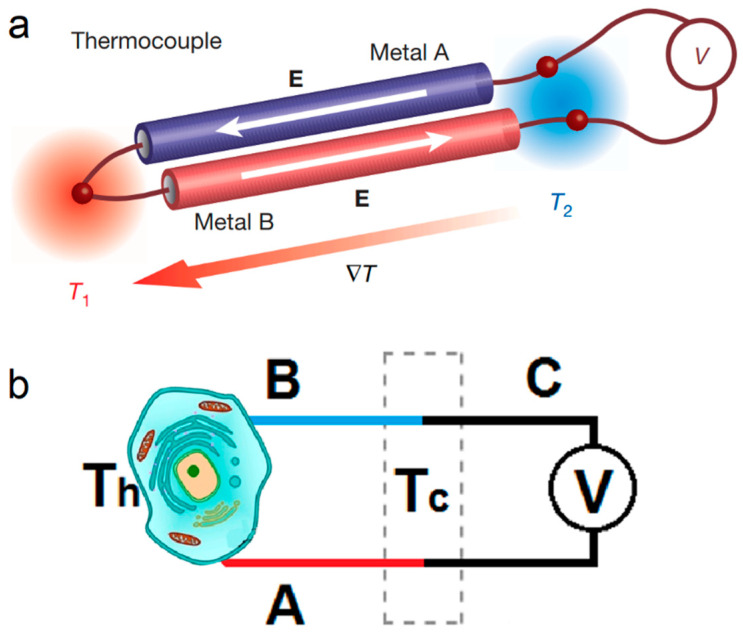
The principle of temperature measurement for the thermocouple. (**a**) The composition of a thermocouple based on Seebeck effects [78]. (**b**) Schematic diagram of the cell temperature measurement using the thermocouple.

**Figure 4 sensors-22-05093-f004:**
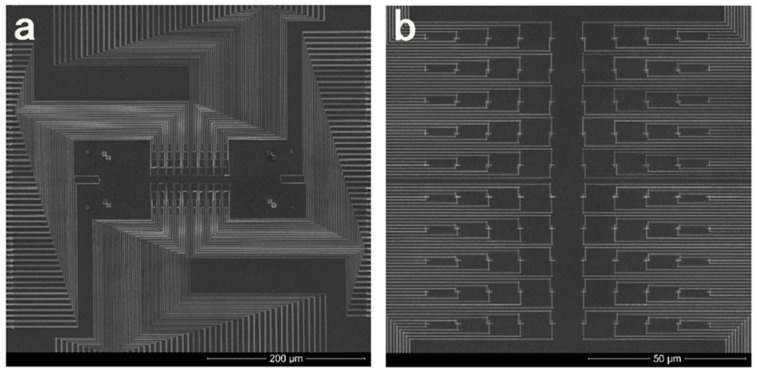
SEM micrographs of a 10 × 10 array of Pr–Cr TFTC on a SiO_2_/Si wafer. (**a**) An overview of the array of 10 × 10 TCs and their 100 leads. (**b**) The central of the 100 TFTCs.

**Figure 5 sensors-22-05093-f005:**
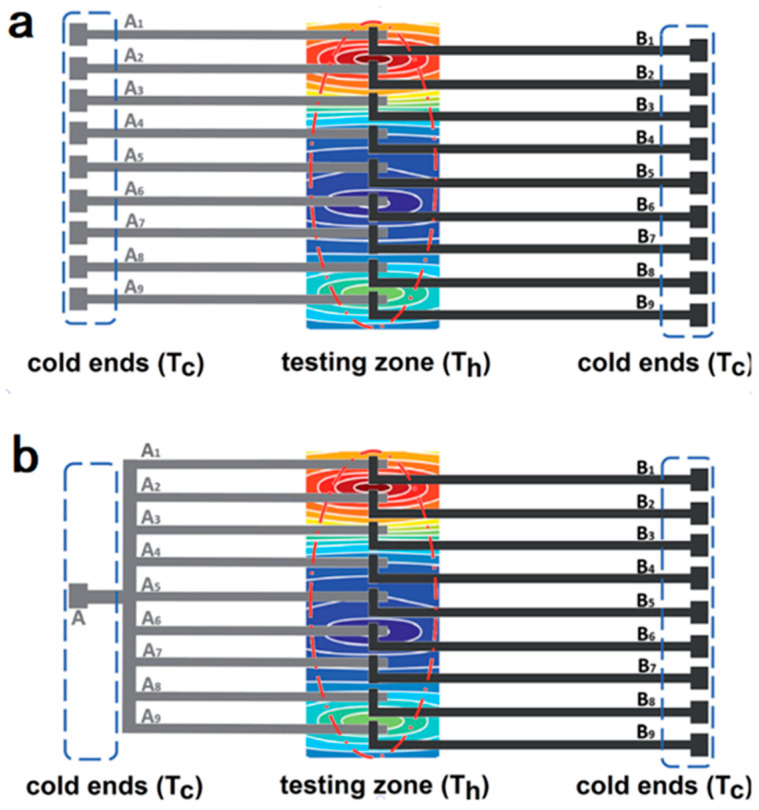
A schematic diagram for saving half leads for a TFTC array [85]. (**a**) Initial design of metal leads in TFTC arrays. (**b**) The principle for saving half of the leads of a TFTC array.

## Data Availability

Not applicable.

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
