# Peer review of "Thermal Probing Techniques for a Single Live Cell"

_sensors, 2022, doi:10.3390/s22145093_

Round 1
Reviewer 1 Report
The authors review the pros and cons of a single live cell's existing thermal probing techniques. Although the review is relatively well written, careful proofreading is necessary and can be improved further. However, the authors may wish to revise the conclusion section, especially since the middle two paragraphs seem a bit redundant; they can shorten it. Overall, the figure quality is good.
Author Response
We thank the reviewer for the professional comments. The English has been polished by a native speaker. In addition, we have shorten the conclusion section to make our point of view clearer. Following the comments of reviewers, the manuscript has been improved. Please review the revised manuscript attached.
Reviewer 2 Report
I quite like the idea of this review article, which is to focus on the development, implementation, and optimization of temperature probes that operate in living cells. Nonetheless, there are several minor issues that combine to detract from my overall enthusiasm for this manuscript, and will need to be addressed before I can recommend publication. These include:
- The authors use the word “metabolization” rather than “metabolism.” They are advised to consider which word is appropriate for their particular context.
- Certain key phrases in the introduction are grandiose and have no place in a technical scientific manuscript. For example, claiming that a full understanding of sub-cell-level processes may “reveal the origins of lives on our planets” is grandiose and should be modified/ removed.
- The technical definition of ‘optical probes’ in the introduction is listed as ‘beams of photons with different wavelengths.’ While this may be accurate, ti is in no way useful to actual researchers who work with optical probes, and use the “photons with different wavelengths” in procedures such as absorption and/or fluorescence emission, including monitoring of optical signal changes. All of the definitions used in the introduction should be reviewed and clarified as needed.
- Also in the introduction, the authors refer to “sophisticated optical equipment” in talking about fluorescence monitoring of the temperature inside living cells. More accuracy around this characterization of ‘sophisticated optical equipment’ should be added to the description.
- In general, this manuscript suffers from a non-trivial number of syntax-based and English language errors that distract from the ability to focus exclusively on the scientific content of the mansucri0pt. The authors should consider how to best address this issue.
- In the introduction, the authors introduce the concept of the “Seebeck Effect” but provide no explanation about the definition of that effect and/or why that effect is useful in this particular context. More information should be provided the first time that the “Seebeck Effect” is introduced.
- In the introduction, the authors state “were an increment of 7.5 K was detected in certain cell process.” This phrase is nonsensical as written and needs to be corrected.
- The authors refer to the fluorophore “NBD” without specifying the full name of the fluorophore. This should be modified so that the full name of the fluorophore is listed the first time, with “NBD” written in parentheses so that it is clear that it is meant to be the abbreviated way of referring to that fluorophore.
- The authors state that “for fluorescent thermometers bearing two or more fluorophores,” that FRET is necessary. This is simply not correct. FRET can be used in situations in which multiple fluorophores are present, but it certainly does not have to be used in those cases. This sentence should be modified to more accurately reflect the scientific reality.
- The authors indicate that the finding that the cell nuclei have a higher temperature than the rest of the cell was “not a surprising founding.” First of all, the correct word here is “finding” rather than “founding.” Second of all, the authors should add information to explain why, in their opinion, this finding was not surprising, so that the appropriate scientific context for this assertion is clearly stated.
- The authors refer to a “cation unit benefit of cell autonomous penetration.” This phrase is nonsensical and needs to be modified.
- In the example in which the cell nuclei were found to be at a higher temperature, the authors use a temperature difference of 1.0 K and 1.0 degrees Celsius interchangeably. For clarity, they should decide on and use only one temperature scale for measurements.
- The authors refer to the fluorescent probe “Mito Thermo Yellow (MTY).” They should include the structure of this fluorescent probe somewhere in the manuscript.
- In that same section, the authors assert that MTY is “resistant to interference from the intracellular environment.” More information in support of this assertion needs to be provided.
- The authors state that fluorescence nano-thermometers “have achieved impressing progresses.” This is a nonsensical phrase and needs to be corrected.
- The authors assert that intracellular fluorescence excitation requires “green, blue, or ultraviolet” wavelength of light. This is a fairly limiting assertion, and I am not sure that there is scientific evidence to support the imposition of such limits. More information needs to be provided.
- The authors state that one solution to the need for a small working distance is “optical systems with an inverted microscope.” This is an ambiguous statement, and more information needs to be provided about what these systems are and specifically how they address the issue of the need for a small working distance effectively.
- When talking about the fabrication of TFTCs, the authors indicate that these have to be fabricated in a clean room, and that “they have the advantages of simple fabrication.” It is unclear how the need for a clean room equates with the characterization of these methods as “simple fabrication.” More clarification is requested.
More broadly, I am confused about how the authors have selected key parts of this broad research area for inclusion in this review article. The article appears to have selected certain examples of temperature sensing inside living cells, without providing a reason or logic that underlies the selection of the aforementioned examples. The authors are advised to critically consider the information they have included, to provide a logic/rationale for the inclusion of these examples and the exclusion of others, and to consider including additional examples that would provide more consistency in the selection and review process.
Reviewer 3 Report
The study of the processes occurring inside living cells has been a priority and most important area of ​​bioscience for many years. Temperature is one of the most important parameters in cell activity, biochemical reactions and metabolism. The presented review describes well the problems in the field of studying living cells and their properties, in particular, temperature. The authors quite well described the state of the art in the field of technologies and approaches to the study of the temperature of single living cells. The paper proposes a justified approach to using a hybrid system for thermal probing of single living cells by combining a stand-alone platform with a thin-film matrix of nanothermocouples and an optical platform for fluorescent nanothermometry. The advantages of this approach are presented, in particular, such a system can be used to measure the absolute local temperature of the tested target cell over a long period of time, up to days and weeks, and from time to time to map the temperature distribution inside the cell on a submicron scale. Further development in the field can open up new horizons in the diagnosis of tumor cells under the influence of various drugs, chemical and physical effects of new methods of treatment.
The presented review is definitely of interest.
Author Response
We appreciate the professional comments. Following the comments of reviewers, the manuscript has been improved. Please review the revised manuscript attached.
Round 2
Reviewer 2 Report
While the authors have done a good job responding to the majority of my comments on the previous version of this manuscript, I am concerned that as part of the revision, they have elected to include a definition of a thermocouple which is taken verbatim from the corresponding Wikipedia page.
In particular, the authors wrote, " A thermocouple is an electrical device consisting of two dissimilar electrical conductors forming an electrical junction.”
The corresponding Wikipedia page states, “A thermocouple is an electrical device consisting of two dissimilar electrical conductors forming an electrical junction.”
I have not done a full check of the rest of the manuscript for similar cut-and-paste phrases/sentences, but I would encourage the editors and authors to take this situation seriously.
Author Response
Thank the reviewer for your comments. As you’ve seen, the definition of the “thermocouple” in the manuscript was indeed from the Wikipedia to avoid misrepresentation, in order to explain the thermocouple more professionally. But we are so sorry that we neglected to cite its source. So we have added the source of the definition, as shown in Fourth Paragraph, Page 1:
A thermocouple is “an electrical device consisting of two dissimilar electrical conduc-tors forming an electrical junction” (Wikipedia).
The rest of the manuscript was written by ourselves, and we have checked it again.
Round 3
Reviewer 2 Report
The authors cannot respond to my previous critique by citing Wikipedia as their reference and leaving the text verbatim.
I recommend that this manuscript be rejected for publication.
Author Response
We accept the reviewer’s critique, and give own interpretation of “thermocouple” in the revised manuscript, as shown in Line 45-49, Page 1-2:
“A thermocouple is usually composed of two electrical conductors with different Seebeck coefficients, serving as a thermometer based on the thermoelectric principle. It produces a temperature-dependent voltage as a result of the Seebeck effect, which is the electromotive force that develops across two points of an electrically conducting material when there is a temperature difference between them [22].”